# Poly(methacrylate citric acid) as a Dual Functional Carrier for Tumor Therapy

**DOI:** 10.3390/pharmaceutics14091765

**Published:** 2022-08-24

**Authors:** Bo Yu, Yiping Shen, Xuejie Zhang, Lijuan Ding, Zheng Meng, Xiaotong Wang, Meihua Han, Yifei Guo, Xiangtao Wang

**Affiliations:** 1Institute of Medicinal Plant Development, Chinese Academy of Medical Sciences & Peking Union Medical College, No. 151, Malianwa North Road, Haidian District, Beijing 100193, China; 2Key Laboratory of Bioactive Substances and Resources Utilization of Chinese Herbal Medicine, Ministry of Education, Chinese Academy of Medical Sciences & Peking Union Medical College, Beijing 100094, China; 3Key Laboratory of New Drug Discovery Based on Classic Chinese Medicine Prescription, Chinese Academy of Medical Sciences, No. 151, Malianwa North Road, Haidian District, Beijing 100193, China; 4Beijing Key Laboratory of Innovative Drug Discovery of Traditional Chinese Medicine (Natural Medicine) and Translational Medicine, No. 151, Malianwa North Road, Haidian District, Beijing 100193, China

**Keywords:** poly(methacrylate citric acid), doxorubicin, pH sensitive, chelating Cu^2+^ effect

## Abstract

Owing to its pH-sensitive property and chelating Cu^2+^ effect, poly(methacrylate citric acid) (PCA) can be utilized as a dual functional nanocarrier to construct a nanodelivery system. Negatively charged carboxyl groups can interact with positively charged antineoplastic drugs through electrostatic interaction to form stable drug nanoparticles (NPs). Through drug experimental screening, doxorubicin (DOX) was selected as the model drug, PCA/DOX NPs with a diameter of 84 nm were prepared, and the drug-loading content was 68.3%. PCA/DOX NPs maintained good stability and a sustained release profile. Cell experiments presented that PCA/DOX NPs could inhibit effectively the growth of 4T1 cells; the IC_50_ value was decreased by approximately 15-fold after incubation for 72 h. The cytotoxicity toward H9C2 was decreased significantly. Moreover, based on its ability to efficiently adsorb copper ions, PCA showed good vascular growth inhibition effect in vitro. Furthermore, animal experiments showed that PCA/DOX NPs presented stronger anticancer effects than DOX; the tumor inhibition rate was increased by 1.5-fold. Myocardial toxicity experiments also confirmed that PCA reduced the cardiotoxicity of DOX. In summary, PCA/DOX NPs show good antitumor efficacy and low toxicity, and have good potential for clinical application.

## 1. Introduction

Polymers with unique advantages [1], including multifunctional properties [2], hydrophobicity [3], biodegradability [4], stealth abilities [5], and active targeting [6], are utilized as biomaterials in tumor treatment [7] and tissue engineering [8]. In cancer treatment, chemotherapeutic drugs present several drawbacks, such as poor aqueous solubility and severe side effects, among others, which limit their bioavailability [9,10]. To overcome these drawbacks, antitumor drugs are loaded onto biomaterials to form nanoscale drug delivery systems [11]. Compared to free drugs, nanodrug delivery systems exhibit the advanced properties, such as increased solubility and bioavailability [12], improved stability [13], prolonged the circulation time [14,15], controlled biodistribution and rate of release [16,17], active targeting [18], and reduced side effects [19], among others. These properties reveal that the nanoscale drug delivery systems (NDDS) may be the ideal delivery system for hydrophobic drugs.

The cellular environment in which tumors or tumor stem cells survive is called the tumor microenvironment (TME) [20,21]. Owing to the abnormal proliferation of tumor cells, the TME has several properties that are different from those of normal tissues [22,23]. The rapid proliferation of tumor cells requires a large amount of oxygen and nutrients, which are transported through blood vessels [24]. Meanwhile, the rapid proliferation produces many metabolites, e.g., lactic acid, inducing TME acidification [25]. Based on these specific characteristics, several strategies are designed to improve antitumor efficacy. Because of the parameters of the TME, pH-sensitive delivery systems have been widely researched [26,27]. To construct pH-sensitive nanoscale drug delivery systems, two strategies have been utilized, chemical conjugation and physical electrostatic interaction [28,29]. The physical electrostatic interaction method exhibits unique properties, including maintaining the therapeutic efficacy of drugs and exhibiting excellent biosafety [30,31]. Among hydrophobic antitumor drugs, doxorubicin (DOX) is a typical cationic drug, which damages DNA by inhibiting the production of topoisomerase II and free radicals [32]. DOX is usually entrapped by anionic nanocarriers to construct a pH-sensitive nanoscale drug delivery system based on electrostatic interaction, and these DOX-loaded nanoparticles present good antitumor activity [33,34]. 

Additionally, avoiding neovascularization is considered a feasible way to treat cancer, because it can block the transport of oxygen and nutrients to the tumor cells [35,36]. Some studies have explored the delivery of antiangiogenic agents to tumor tissues to optimize antitumor efficacy [36,37]. It has been reported that copper ions present the ability to initiate an angiogenic response, and the concentration of copper in many cancer tissues and sera is higher than that in normal tissues [38,39]. Therefore, decreasing the concentration of Cu^2+^ ions might inhibit angiogenesis in tumor tissues, inducing tumor cell apoptosis. Nanomaterials with chelation abilities can reduce the Cu^2+^ concentration, and thus have potential application as antiangiogenic agents.

Poly(methacrylate citric acid) (PCA) is a new anionic nanomaterial that was polymerized from citric acid methacrylate monomer in our previous study [40]. Based on the abundant carboxyl groups in citric acid, PCA is highly negatively charged. Additionally, PCA presents excellent biosafety and biocompatibility, and can efficiently chelate copper ions. Given these results, it seems that the anionic polymer PCA, due to its pH sensitivity and chelation effects, can be utilized as a dual functional nanocarrier, to promote antitumor efficacy. NDDS are usually prepared from amphiphilic block copolymers [41,42]. Thus, there is some uncertainty as to whether PCA, as a hydrophilic polymer, can be used to successfully construct NDDS. It has been reported that hydrophilic PEG derivatives can be used as nanocarriers to prepare stable, high-drug-loading NDDS, which can effectively improve the antitumor effect of the drug [43]. Therefore, a similar preparation method can be utilized to construct NDDS with the hydrophilic PCA as the nanocarrier.

To promote anticancer activity, in this study the hydrophilic anionic polymer PCA was used as the drug carrier and loaded with DOX to construct the nanodrug delivery system, PCA/DOX nanoparticles (NPs), via electrostatic interactions, which was expected to accomplish two goals, pH-sensitive DOX delivery and Cu^2+^ concentration reduction in tumor tissues. After successfully preparing the nanoparticles, their physicochemical properties, such as drug-loading content, entrapment efficacy, particle size, and morphology were estimated, and then their pH-sensitive release profiles, antiangiogenic activity, antitumor efficacy, and toxicity were investigated.

## 2. Materials and Methods

### 2.1. Materials

Poly (methacrylate citric acid) (PCA, Mn = 1.57 × 10^4^) was prepared by free radical polymerization, according to the method described in our previous article [40]. Doxorubicin (DOX), resveratrol (RES), nifedipine (NIF), ibuprofen (IBU), hydroxycamptothecin (HCPT), celastrol (CSL), honokiol (HK), and podophyllotoxin (POD) were purchased from Aladdin, Shanghai, China. Fetal bovine serum, penicillin G, and streptomycin were purchased from Gibco, Grand Island, NY, USA. RPMI 1640 medium and phosphate-buffered saline (PBS) were purchased from HyClone, Logan, UT, USA. ECGM endothelial cell culture medium was purchased from Roles-Bio, Guangzhou, China. A 24-well sterile culture plate and a 96-well sterile culture plate were purchased from Corning, Corning, NY, USA. HPLC-grade methanol was purchased from Fisher Scientific, Waltham, MA, USA. Other reagents and solvents were purchased at reagent grade and could be used without further purification.

### 2.2. Animals and Cell Line

The 4T1 murine breast cell line, H9C2 cell line, and HUVEC cell line were purchased from the National Infrastructure of Cell Line Resource (Beijing, China). The 4T1 cell line was cultured in RPMI-1640 medium, the H9C2 cell line was cultured in DMEM medium, and the HUVEC cell line was incubated in ECGM medium. All of the media were supplemented with 10% fetal bovine serum, 100 units mL^−1^ of penicillin G and streptomycin. The culturing was carried out at 37 °C with a 5% CO_2_ atmosphere.

BALB/c mice (20 ± 2 g) were ordered from Vital River Laboratory Animal Technology Co., Ltd. (Beijing, China), and raised in an SPF laboratory for 1 week. The mice were fed a standard diet of food and water. The animal experiments were approved by the Animal Ethics Committee of Peking Union Medical College (Beijing, China). All operations were conducted in accordance with the Guidelines and Policies for Ethical and Regulatory for Animal Experiments. The ethical approval number was SLXD–20190423015.

### 2.3. Preparation of Drug-Loaded PCA Nanoparticles

Hydrophobic drugs (20 mg) and PCA (5 mg) were dissolved in DMF (1 mL), and the mixture was added dropwise into deionized water (5 mL) under ultrasonic conditions. Subsequently, the system was transferred to dialysis bags (MWCO 8000–14,000 Da) and dialyzed with deionized water for 4 h (2 L/h). A drug-loaded nanoparticle (NP) solution was obtained after homogenization at 1600 bar pressure for 5 times. 

### 2.4. Particle Diameter and Morphology

The size distribution, polydispersity (PDI), and zeta potential of the samples were determined using a Malvern Zetasizer 3000 system (Malvern Instruments Ltd., Malvern, UK), and the concentration was 2 mg/mL. 

The PCA/DOX NPs solution (40 μg/mL) was dropped on a clean silica gel sheet to dry, and fixed with conductive glue. After being sprayed with gold for 6 min under negative pressure, the samples were observed by a scanning electron microscope (SEM) at a current of 30 mA and a voltage of 30 mV.

### 2.5. Drug-Loading Content (DLC) and Encapsulation Efficiency (EE)

A total of 5 mL of PCA/DOX NPs solution was lyophilized to obtain NPs powder. After precise weighing, the NPs powder was dissolved in 1 mL of methanol, and vortexed for 15 min. The NPs methanol solution was centrifuged at 13,000 r/min for 30 min, and the supernatant was collected, diluted, and detected by HPLC-UV analysis. The measurement was conducted on a C18 column (250 × 4.6 mm^2^) at 25 °C with the eluent of methanol and 0.2% phosphoric acid (50/50, *v*/*v*), and the wavelength was set to 266 nm. The flow rate was 1.0 mL/min, and the injection volume was 20 μL. The DLC and EE were calculated as follows:DLC (%) = A/B × 100%;(1)
EE (%) = A/C × 100%;(2)
where A is the weight of DOX in the PCA/DOX NPs, B is the weight of the PCA/DOX NPs, and C is the total DOX weight.

### 2.6. Stability of NPs

The PCA/DOX NPs solution (4 mg/mL) was sealed and stored at 4 °C, and samples were taken at preset time points (day 0, 2, 4, 6, 8, 10, 12, 14, and 28). The size distribution of NPs was determined using a Malvern Zetasizer 3000 system and was measured 3 times in parallel.

The PCA/DOX NPs solution (4 mg/mL) was mixed with 10% glucose solution at 1:1 equal volume to form an isotonic solution. Meanwhile, the PCA/DOX NPs solution was mixed with mouse plasma at a ratio of 1/4 (*v*/*v*). All of these mixtures were incubated at 37 °C; the particle size and potential of samples were measured at 0, 2, 4, 6, 8, 10, 12, and 24 h, and each sample was measured 3 times in parallel. 

### 2.7. Study on the Release of PCA/DOX NPs In Vitro

The PCA/DOX NPs solution (1 mg/mL, 2 mL) and DOX solution (1 mg/mL, 2 mL) were placed in dialysis bags (MWCO 8000–14,000 Da), which were dropped into 50 mL of release medium (PBS solution containing 1% SDS, pH = 7.4, 5.5), and stirred at 37 °C (100 r/min). Release medium (1 mL) was withdrawn at the set time points (0.25, 0.5, 1, 2, 4, 6, 8, 10, 12, 24, 48, 72, 96, 120, and 144 h), and then the same amount of fresh PBS was added. The concentration of DOX was determined by HPLC according to the above chromatographic conditions. The cumulative release rate of DOX was calculated and the cumulative release curve was drawn. Each sample was tested in parallel in triplicate.

### 2.8. Cytotoxicity Assay

At 37 °C and 5% CO_2_, 4T1 cells and H9C2 cells in the logarithmic growth phase were seeded in a 96-well plate (8 × 10^3^ cells/well). The samples were diluted with fresh RPMI-1640 medium to achieve concentrations of 0.01, 0.1, 0.5, 1, 2, 5, 10, 20, and 100 μg/mL. The DOX solution and PCA/DOX NPs with different concentrations were added in wells (150 μL per well, six wells for each sample) after 24 h incubation.

After culturing for 48 or 72 h, 10 µL of CCK-8 solution was added to each well and incubated for another 1.5 h. The optical density (OD) value was measured with an ELISA microplate reader at 450 nm. Cell inhibition rate was calculated as follows, and the IC_50_ was calculated by GraphPad prism 5 software: Cell_inhibition rate_ (%) = (1 − OD_treated_/OD_blank_) × 100%.(3)
OD_treated_ is the OD value of cells treated with DOX solution and PCA/DOX NPs samples, and OD_blank_ is the OD value of cells grown in fresh RPMI-1640 medium.

### 2.9. Vascular Growth Inhibition Experiment

The BD Matrigel matrix gel and ECGM medium were mixed at a ratio of 9/1 (*v*/*v*), and the mixed matrix gel (250 μL) was slowly plated in a 24-well plate. Then, it was stored at 4 °C for 10 min to flatten the gel surface and then hardened at 37 °C for 2 h. The HUVEC cells (5 × 10^5^) were seeded on solidified Matrigel and treated with blank ECGM medium (blank control), ECGM medium containing Cu^2+^ ions (25 μmol/mL, positive control), ECGM medium containing Cu^2+^ ions (25 μmol/mL), and PCA (100 μmol/mL). The cells were photographed at 4, 18, and 24 h to compare the tube formation. The images were recorded under the same scale bar.

### 2.10. Investigation of Antitumor Efficacy and Systemic Toxicity

The animal model was established by the subcutaneous injection of 4T1 cell suspension (0.2 mL, 1 × 10^7^ cells/mL) into the right armpit of BALB/c mice. When the tumor volume exceeded 150 mm^3^, the mice were randomly divided into 4 groups (10 animals per group), and all of them were treated via tail vein injection. The negative control group was administered normal saline; the positive control group was administered DOX solution (3 mg/kg, DOX equivalent concentration); the blank control group was administered PCA solution at a concentration of 0.75 mg/kg; and the test group was administered PCA/DOX NPs (3 mg/kg, DOX equivalent concentration).

All mice were administered 0.2 mL of the sample solution every two days 7 times, and the first administration was recorded as day 0. The body weight and tumor volume of the mice were monitored (tumor volume= (length × width^2^)/2) every two days during the entire procedure. After measuring tumor size, body weight, and collecting blood, all mice were sacrificed on day 14. The tumor tissue and main organs (heart, liver, and spleen) were removed and weighed. The tumor inhibition rate (TIR), liver index (LI), and spleen index (SI) were calculated as follows:TIR (%) = (1 − average tumor weight of treatment group/average tumor weight of negative control group) × 100%.(4)
Heart index (%) = (heart weight/body weight) × 100%.(5)
Liver index (%) = (liver weight/body weight) × 100%.(6)
Spleen index (%) = (spleen weight/body weight) × 100%.(7)

### 2.11. Biochemical Parameters

Serum was obtained via centrifugation (5000 rpm, 5 min) after blood collection. Enzyme biomarkers, including lactate dehydrogenase (LDH), aspartate aminotransferase (AST), phosphocreatine kinase (CK), and phosphocreatine kinase isoenzyme (CK-MB) in the serum were detected by automatic biochemical instruments. The measurement was conducted in quintuplicate.

### 2.12. Histological Assessment 

The tumor and heart tissues of mice were fixed in 10% formalin solution, and 5 mm slices were prepared. Then, the tissue specimens were stained with hematoxylin–eosin for 5 min. The morphology of the slices was observed and photographed using a light microscope (Eclipse E100, Nikon, Tokyo, Japan).

### 2.13. Statistical Analysis

The statistical analyses of experimental data was calculated by independent-sample *t*-tests using IBM SPSS Statistics software, Version 21 (IBM Corporation, Armonk, NY, USA).

## 3. Results and Discussion

### 3.1. Preparation of Drug-Loaded Nanoparticles

Poly(methacrylate citric acid) (PCA) was synthesized successfully. The molar mass was 1.57 × 10^4^, and the polydispersity index was 1.29. PCA showed excellent aqueous solubility, the hydrodynamic diameter was 379.3 ± 14.6 nm, and the zeta potential was −51.7 mV. To estimate the applicability of hydrophilic PCA as a nanocarrier, several hydrophobic drugs, including doxorubicin (DOX), resveratrol (RES), nifedipine (NIF), ibuprofen (IBU), hydroxycamptothecin (HCPT), celastrol (CSL), honokiol (HK), and podophyllotoxin (POD) were selected to construct drug-loaded nanoparticles (NPs). These hydrophobic drugs were selected according to their zeta potential (Appendix A), which can be divided into two groups: one is the positively charged group, including RES, NIF, IBU, and DOX, and the other is the negatively charged group, including HCPT, CSL, HK, and POD. The antisolvent precipitation method combined with ultrasonication was utilized to prepare drug-loaded nanoparticles. These hydrophobic drugs and PCA were dissolved in DMF to form an organic phase, and then were injected into deionized water. During this procedure, hydrophobic drugs are physically encapsulated by PCA via electrostatic interactions, hydrophobic interactions, hydrogen bonds, etc.

After homogenization, NIF, HK, and POD presented significant precipitation; RES, IBU, DOX, HCPT, and CSL showed a homogeneous emulsion (Appendix A). However, after storage overnight, RES and IBU presented precipitation, revealing that these systems are unstable (Appendix A). When the storage time was 48 h, the appearance of DOX, HCPT, and CSL emulsion showed no change (Figure 1a); when further prolonging the storage time to 7 days, no significant change was shown. It seems that DOX, HCPT, and CSL can be loaded successfully to form a stable nanodrug delivery system with PCA as a nanocarrier. HCPT and CSL can form carrier-free nanocrystals in accordance with the findings of previous studies [44,45]. Hence, it is difficult to conclude that HCPT and CSL can be entrapped by PCA. These results suggest that of these model drugs, only DOX can be entrapped by PCA effectively to construct stable drug-loaded NPs. This phenomenon may be explained by the electrostatic interactions between drugs and nanocarriers. PCA, as a poly acid, presents a negative charge of −51.7 mV, which can interact with the positively charged drug DOX (Figure 1b). Although RES, NIF, and IBU also have a positive charge, the absolute value is low, resulting in weak electrostatic interactions; DOX shows a high zeta potential, inducing strong electrostatic interactions to form stable DOX-loaded nanoparticles. In summary, DOX can be physically entrapped by PCA to aggregate stable PCA/DOX NPs.

This result proves that it is possible to construct a drug delivery system with a hydrophilic polymer and a hydrophobic drug via electrostatic interactions. Next, the physicochemical properties, stability, and antitumor efficacy of PCA/DOX NPs were measured in detail.

### 3.2. Characterization of PCA/DOX NPs

The particle size of the PCA/DOX NPs was 84.4 nm, and their polydispersity index (PDI) was 0.19. Figure 1c shows the particle size distribution curve. It is clear that NPs with a diameter of less than 100 nm can accumulate effectively in tumor tissue via enhanced vascular permeability and retention (EPR) effects [46,47]. Thus, PCA/DOX NPs might exhibit a good accumulation rate in tumor tissue. The PDI was less than 0.3, indicating that PCA/DOX NPs could exist in the dispersion system uniformly and stably. Moreover, the particle diameter of PCA/DOX NPs was smaller than those of other drug-loaded nanoparticles, which was almost 170–310 nm [43,48]. It is possible that the electrostatic interaction between PCA and DOX induced the more compact nanostructure and the smaller diameter. A scanning electron microscope observation image is shown in Figure 1d. PCA/DOX NPs presented an irregularly spherical morphology and relatively uniform sizes. The particle size is approximately 70 nm. These results reveal that PCA/DOX NPs existed in a compact nanostructure.

After detecting by HPLC, the DLC and EE were calculated based on the actual DOX concentration, which was 68.3% and 85.4%, separately. High DLC and EE levels could be explained by the high density of carboxyl groups in PCA, which present more active sites for interaction with the amine groups in DOX. The high DLC could enhance the biosafety and reduce the side effects owing to the reduction in administration times.

### 3.3. Stability of PCA/DOX NPs

It can be seen that PCA/DOX NPs have good storage stability for 28 days; the particle size distribution curves are shown in Appendix A. Within the initial 7 days, the average particle diameter increased slightly and stabilized at 110 nm (Figure 2), which could be attributed to Ostwald ripening. When prolonging the detection time to 28 days, the particle sizes were maintained at approximately 110 nm. When investigating the stability of the media, it was found that PCA/DOX NPs remained stable in plasma and 5% glucose solution. Although the particle size was enlarged slightly to 157 and 175 nm in plasma and 5% glucose solution, respectively, no turbidity or coagulation phenomena appeared during the entire procedure. These results could be attributed to the weakened electrostatic interaction that was affected by glucose molecules and protein, which slightly loosened the compact structure of PCA/DOX NPs. This result indicated that PCA/DOX NPs could be dissolved in glucose solution for intravenous administration and presented good stability during the blood circulation procedure.

### 3.4. Drug Release In Vitro

Electrostatic interaction can be affected by the environmental pH. The in vitro release curves of PCA/DOX NPs in different buffer solutions were measured to study the effect of pH (Figure 3). Compared with free DOX, PCA/DOX NPs showed a sustained release property. The total release of free DOX was accomplished within 24 h, and a cumulative release rate of 84.4% was achieved at pH 7.4, while, only 18.4% DOX was released from PCA/DOX NPs at pH 7.4, and the release procedure could be sustained for at least 6 days. This might be caused by the electrostatic interaction between PCA and DOX. It has been reported that the pKa values of citric acid are 3.1, 4.8, and 6.4 [49]; the pKa value of DOX was 8.3 [50]. Citric acid is completely deprotonated at pH 7.4, and it can interact with DOX to form stable NPs. Additionally, the structure and steric hindrance of NPs hampered DOX diffusion. Therefore, slow DOX release from PCA/DOX NPs was observed. Moreover, PCA/DOX NPs were released more quickly in an acidic environment. The cumulative release rate at pH 5.5 was 38.2% within initial 24 h and two times higher than that at pH 7.4, indicating that PCA/DOX NPs are sensitive to the acidic tumor microenvironment.

Additionally, dynamic release models for PCA/DOX NPs were constructed. When DOX release from PCA/DOX NPs was performed at pH 7.4, the dynamic release model fits the Ritger–Peppas equation; the n value (0.57) is between 0.45 and 0.89. On the contrary, the dynamic release model of PCA/DOX NPs at pH 5.5 coincides with the Higuchi equation. These model equations are shown in Appendix A. The different dynamic release model reveals that PCA/DOX NPs present different release profiles dependent on the pH value.

The pH sensitive release profile of DOX could be attributed to high solubility in acidic solution owing to amine groups of DOX being protonated. Furthermore, the intensity of electrostatic interactions between PCA and DOX was affected by the pH value. Under physiological conditions (pH 7.4), citric acid was deprotonated and interacted with DOX via strong electrostatic interactions. However, when the pH decreased to 5.5, a partial carboxyl group was protonated and the electrostatic interaction was weakened. Therefore, accelerated DOX release was observed at pH 5.5. These results suggest that PCA/DOX NPs can achieve efficient intracellular DOX release in tumor tissue. 

### 3.5. Cytotoxicity Assay

The CCK-8 method was utilized to investigate the cytotoxicity of PCA/DOX NPs in vitro, and DOX injection was used as the control. The biosafety of PCA toward normal cell lines, including HUVEC and H9C2 cell lines, was studied first (Appendix A). After coincubation for 48 h, the cell viability of PCA was over 90% for both normal cell lines, indicating the good biosafety of PCA.

Next, the cytotoxicity of PCA/DOX NPs toward the 4T1 tumor cell line and normal H9C2 cell line were researched in detail. Concerning the 4T1 cell, DOX and PCA/DOX NPs both exhibited concentration-dependent and time-dependent phenomena. The cell inhibition rate was promoted significantly when increasing the concentration of DOX and prolonging the coincubation time. Additionally, PCA/DOX NPs presented higher cell inhibition rates than free DOX. After incubating for 48 h, the IC_50_ value was 8.73 and 1.76 μg/mL for the free DOX and PCA/DOX NPs (DOX equivalent concentration), respectively (Figure 4a). The cell inhibition rate of PCA/DOX NPs was enhanced approximate five-fold comparing with free DOX (*p* < 0.001). Extending coincubation time to 72 h, the cell inhibition rate was further enhanced. The IC_50_ value decreased to 3.78 and 0.26 μg/mL for free DOX and PCA/DOX NPs (DOX equivalent concentration) correspondingly (Appendix A), and the cell inhibition rate of PCA/DOX NPs was enhanced approximately 15-fold when compared with free DOX (*p* < 0.001). When the coincubation time changed from 48 to 72 h, the cell inhibition rate increased 6.8-fold. These results are consistent with previous reports [51,52,53]. The possible mechanism for nanoparticles with enhanced antitumor effect is that they are internalized into tumor cells through the endocytosis pathway, while free drugs pass the cell membrane through passive diffusion.

Furthermore, to study the cytotoxicity toward normal cardiac cells, the H9C2 cells were treated with PCA/DOX NPs and incubated for 48 h, and a DOX injection was utilized as a positive control (Figure 4b). With the increase in the amount of DOX, the viability of cells treated with free DOX decreased sharply. When the concentration of DOX reached 0.40 μg/mL, no H9C2 cells survived. The IC_50_ of a DOX injection is 0.19 μg/mL. These results suggest that a DOX injection exhibits significant cytotoxicity toward cardiac cells. In contrast, the cytotoxicity of PCA/DOX NPs showed a different tendency. Although the cell viability was also decreased when increasing the concentration of PCA/DOX NPs (DOX equivalent concentration), no dramatic reduction tendency was observed. When the concentration of PCA/DOX NPs was 100 μg/mL, the cell viability of H9C2 exceeded 60%; hence, the IC_50_ could not be calculated according to these data. This phenomenon indicated that the cytotoxicity of PCA/DOX NPs toward normal cardiac cells was lower than that of free DOX.

The enhanced cytotoxicity toward tumor cells and decreased cytotoxicity toward normal cardiac cells could be explained by the pH-sensitive release profiles of PCA/DOX NPs. It has been reported that the concentration of free DOX accumulated in the nucleus is a key parameter that indicates its cytotoxicity, but that DOX NPs in the cytoplasm could not affect the cytotoxicity [54,55]. PCA/DOX NPs may experience intracellular uptake by both tumor and normal cells via the facilitated endocytosis mechanism; after entering the cell membrane, the pH-sensitive release profile induces different DOX concentrations and results in different cytotoxicity levels. H9C2 cells as the normal cardiac cells presented the normal physical pH value (~7.4), and the DOX release rate was slow owing to the strong electrostatic interaction between PCA and DOX. Hence, most of the DOX was encapsulated in PCA/DOX NPs, and the cytotoxicity toward H9C2 cells was decreased significantly. In contrast, the 4T1 cells as the tumor cells showed a relatively low pH value (5.5–6.8), resulting in rapid DOX release and more DOX molecules entered the nucleus. Therefore, enhanced cytotoxicity was exhibited in these cells. 

### 3.6. Vascular Growth Inhibition Experiment

Tumor growth and metastasis are related to angiogenesis, and antiangiogenic therapy is expected to become a novel and effective strategy. It has been reported that copper ions are important for tumor growth. Copper ions exhibit angiogenic effects [38,56]. According to our study, PCA could adsorb copper ions effectively; hence, it should exhibit antiangiogenic activity. As a dual functional drug delivery system, it was expected that PCA could deliver hydrophobic DOX to the tumor tissue effectively via electrostatic interactions. After DOX release, blank PCA may reduce Cu^2+^ ions concentrations via chelation, thus hampering vascular growth. To estimate the antiangiogenic activity of PCA, HUVEC cells were cultured with fresh media containing Cu^2+^ ions and PCA for 24 h. Meanwhile, HUVEC cells cultured with blank ECGM medium (blank control) and ECGM medium containing Cu^2+^ ions (positive control) were studied under the same conditions (Figure 5). As the blank control, the HUVEC cells were cultured with blank ECGM medium, and the angiogenic phenomenon was observed after 24 h incubation (Figure 5, “blank” row, marked by the red dotted line), which could be explained by the culture media containing trace Cu^2+^ ions. After adding Cu^2+^ ions (25 μmol/L) and incubation for 18 h, the phenomenon of endothelial tubule formation on Matrigel was shown in the positive control group (Figure 5, “Cu^2+^“ row, marked by the red dotted line). When extending the incubation time to 24 h, this phenomenon was more significant. Compared with the blank control, the time taken to form endothelial tubules in the positive group was shortened, and the phenomenon of angiogenesis is easier to observe, indicating the angiogenic effect of Cu^2+^ ions. Meanwhile, after adding Cu^2+^ ions (25 μmol/L) and PCA (100 μmol/L), the tubule formation phenomenon was not shown in the whole incubation procedure (Figure 5, “Cu^2+^/PCA” row). At the end of the incubation period, both the blank control and positive control presented complete tube formation, while this phenomenon was not shown in the PCA group. The tube formation rates for blank control, positive control, and PCA group were 100%, 100%, and 0%, respectively. These results were attributed to the chelation effect between PCA and Cu^2+^ ions, resulting in a reduction in Cu^2+^ concentration, revealing that PCA presents potent antiangiogenic activity in vitro owing to its strong copper chelation ability.

### 3.7. Antitumor Efficacy

The in vivo antitumor efficacy of PCA/DOX NPs was estimated via a 4T1-tumor-bearing mice model where a 5% glucose solution (Glu) was utilized as the negative control, a DOX injection (DOX) was utilized as the positive control, and a PCA solution (PCA) was utilized as the blank control. The average tumor volume was 2150, 1374, 996, and 542 mm^3^ for the Glu group, PCA group, DOX group, and PCA/DOX NPs group, respectively (Figure 6a). DOX and PCA/DOX NPs presented a significant inhibitory effect (** *p* < 0.01, vs. Glu group). Based on tumor volume, the tumor inhibition rate was 53% for DOX and 75% for PCA/DOX NPs. Additionally, compared with the DOX group, the tumor volume of the PCA/DOX NPs group was significantly decreased (^#^ *p* < 0.05). 

The average weight of the tumor tissue was 0.64, 0.98, 1.62, and 1.80 g for PCA/DOX NPs, DOX, PCA, and Glu groups, respectively (Figure 6b). The TIR was then calculated using the tumor weight. DOX and PCA/DOX NPs presented moderate to good antitumor activity. The tumor inhibition rate of the DOX group was 45%, and that of PCA/DOX NPs was 65%. Compared with DOX, PCA/DOX NPs emerged with better antitumor efficacy (^#^ *p* < 0.05). These results indicated that PCA/DOX NPs exhibit good antitumor effects.

HE staining was utilized to observe the necrosis of the tumor tissue, and the results are shown in Figure 6c,d. From these images, the necrosis can be seen in both the DOX injection and PCA/DOX NPs group (marked by a black dotted line), indicating that both DOX injection and PCA/DOX NPs presented antitumor activity. Moreover, a larger necrosis area was shown in the PCA/DOX NPs group, revealing that PCA/DOX nanoparticles exhibited the higher antitumor efficacy.

However, PCA showed a slight inhibitory effect of approximately 36% and 10%, based on tumor volume and tumor weight, respectively, but there was no statistical difference between the two (*p* > 0.05, vs. Glu group). The blank PCA presented a particle size of approximately 379 nm, which was larger than that of PCA/DOX NPs. The accumulation rate of PCA in tumor tissues was low owing to the large particle size inducing low permeation. Moreover, angiogenic effect were observed in the early stage of solid tumor formation [57]. After PCA was injected into the mice, the neovascularization in tumor tissue had been formed successfully; hence, antiangiogenic activity of PCA was not shown. This result suggested that to evaluate the antiangiogenic activity, blank PCA should be optimized to obtain the small particle size and administered at the early stage of solid tumor formation [58].

### 3.8. DOX Toxicity Test

#### 3.8.1. Systemic Toxicity Test

To investigate the systemic toxicity of PCA/DOX NPs, the body weight was measured every two days (Figure 7a). The body weight of the Glu group and the PCA/DOX NPs group showed no significant changes. The PCA group showed an increased body weight, while the DOX group presented with a decreased body weight. These results indicate that DOX presents strong systemic toxicity and that PCA exhibits good biological safety. Compared with the DOX group, the body weight of the mice in the PCA/DOX NPs group was significantly different (^##^ *p* < 0.01), but compared with the Glu group, no significant difference were observed (*p* > 0.05). This phenomenon reveals that PCA/DOX may reduce the side effects of DOX when used as a NDDS.

Then, the liver and spleen indexes of the mice were estimated. Compared with the Glu group, the liver/spleen index of the DOX group was decreased significantly, especially the spleen index, which was decreased approximately 2.2-fold (** *p* < 0.01). On the contrary, the liver index and spleen index of the PCA and PCA/DOX NPs groups showed similar value to those of the Glu group; no significant difference was shown (*p* > 0.05). These results confirm the good biological safety of PCA and suggest that PCA can reduce the toxic effects of DOX.

#### 3.8.2. Cardiotoxicity

It is well known that DOX exhibits severe toxic effects on the heart; therefore, the heart toxicity of PCA/DOX NPs was further evaluated. The heart index was calculated based on the heart weight and body weight of the mice (Figure 8a). The value of the heart index was 0.55, 0.40, 0.51, and 0.55 for the Glu group, DOX group, PCA group, and PCA/DOX NPs group, respectively. The heart index of the DOX groups was decreased significantly (* *p* < 0.05, vs. Glu group), while neither the PCA group nor the PCA/DOX NPs group presented significant differences. 

To understand the myocardial injury, myoenzymatic biomarkers including LDH, AST, CK, and CK-MB of the 4T1 tumor-bearing mice were detected. The level of these biomarkers was increased significantly after myocardial injury (Figure 8b) [59]. Compared with the Glu group (negative control), all the biomarkers in the DOX group were increased, and the LDH level exhibited a highly significant difference (*** *p* < 0.001), indicating severe heart injury. On the contrary, the LDH, CK, and CK-MB levels of the PCA/DOX NPs group were increased slightly compared with the Glu group, but decreased significantly compared with the DOX group (^##^ *p* < 0.01 for LDH and ^###^ *p* < 0.001 for CK and CK-MB), revealing PCA/DOX NPs could significantly reduce the myocardial injury effect of DOX.

In addition, when comparing the observation results of heart tissue slices (Figure 8c), heart slices from the Glu group and the PCA group showed no obvious pathological changes. Heart slices from the DOX group had interstitial edema (marked by the black arrow) and the cardiomyocytes presented obvious nucleus lysis and apoptosis (marked by the red arrow), while the heart slices from the PCA/DOX NPs group had slight interstitial edema. Symptoms such as atrophy were less severe than those in the DOX group. 

All these results confirmed that PCA/DOX NPs can significantly reduce the cardiotoxicity of DOX.

## 4. Conclusions

In this study, a new material poly(methacrylate citric acid) (PCA) was used as a carrier for the preparation of drug-loaded nanoparticles. PCA with anions can deliver cationic drugs to tumor tissue via electrostatic interaction. Doxorubicin (DOX) was selected as a model drug to prepare PCA/DOX nanoparticles (NPs) with a high drug-loading content. PCA/DOX NPs showed an irregularly spherical morphology and particles were relatively uniform and small in size. PCA/DOX NPs exhibited good storage stability and remained stable in 5% glucose solution and plasma. The sustained release profile of NPs was shown in an in vitro release evaluation, and rapid DOX release was observed at pH 5.5 owing to the pH-sensitive activity. A cytotoxicity assay showed that PCA/DOX NPs could effectively inhibit the growth of 4T1 tumor cells. The IC_50_ value was decreased significantly, and the cytotoxicity toward H9C2 cells was low. PCA presented the ability to inhibit tumor neovascularization via a reduction in the Cu^2+^ concentration. Animal experiments revealed that PCA could enhance the antitumor effect and reduce the toxicity of DOX; the antitumor activity of PCA/DOX NPs was enhanced 1.5-fold compared to free DOX, and the systemic toxicity and cardiotoxicity were decreased significantly. In summary, PCA/DOX NPs presented good antitumor activity owing to their high drug-loading capacity, small particle size, pH-sensitive release, and chelating Cu^2+^ effect, which suggests their potential application in an anticancer drug delivery system.

## Figures and Tables

**Figure 1 pharmaceutics-14-01765-f001:**
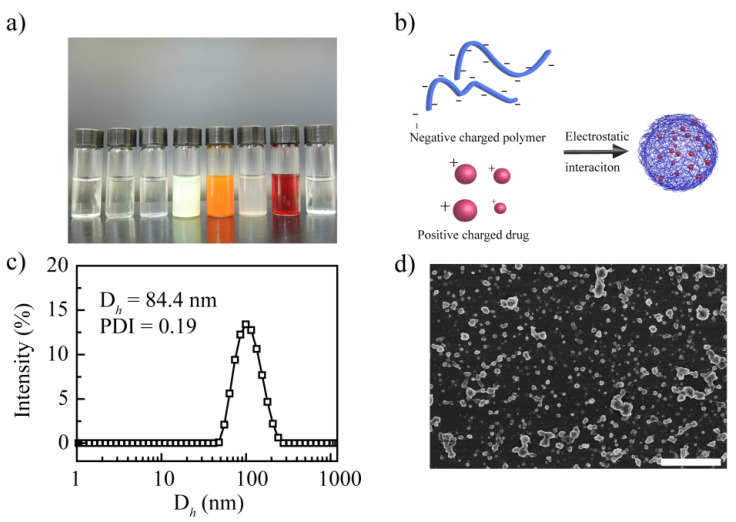
Synthesis and characterization of the prepared drug-loaded nanoparticles. (**a**) Images of the drug-loaded nanoparticles from left to right: RES, NIF, IBU, HCPT, CSL, HK, DOX, and POD. (**b**) Preparation mechanism of PCA/DOX NPs. (**c**) Particle size distribution curve of PCA/DOX NPs. (**d**) SEM image of PCA/DOX NPs. Scale bar 500 nm.

**Figure 2 pharmaceutics-14-01765-f002:**
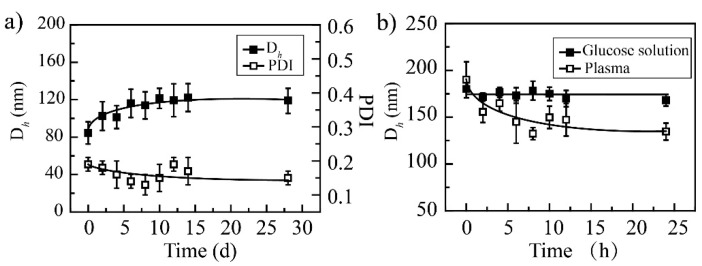
The stability of PCA/DOX NPs: (**a**) storage stability at 4 °C; (**b**) media stability at 37 °C, n = 3.

**Figure 3 pharmaceutics-14-01765-f003:**
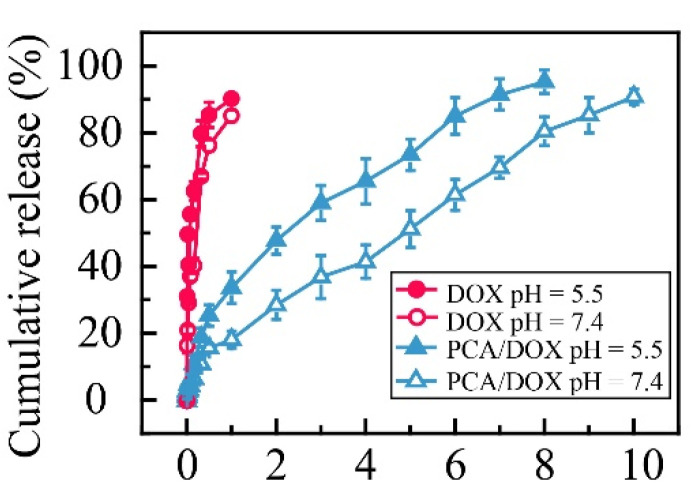
Cumulative release curves of PCA/DOX NPs and DOX in PBS (pH 7.4 and 5.5) at 37 °C (pH 7.4 imitates the normal human environment, and pH 5.5 imitates the acidic environment of the tumor site), n = 3.

**Figure 4 pharmaceutics-14-01765-f004:**
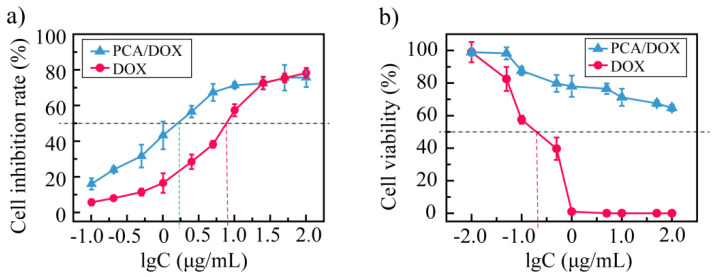
Cytotoxicity investigation results against 4T1 tumor cell line (**a**) and normal H9C2 cell line (**b**) at 37 °C after coincubation for 48 h, n = 5.

**Figure 5 pharmaceutics-14-01765-f005:**
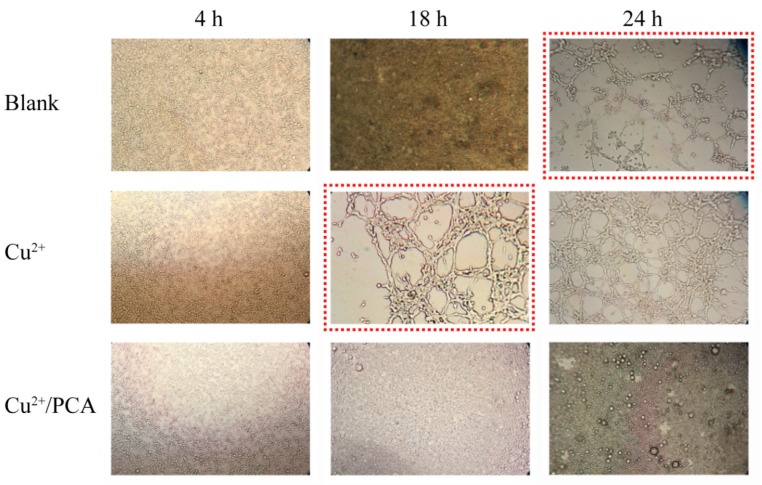
Vascular growth inhibition experiment: HUVEC cells culturing with blank ECGM medium, ECGM medium containing Cu^2+^ ions (25 μmol/L), and ECGM medium containing Cu^2+^ ions (25 μmol/L) and PCA (100 μmol/L) on Matrigel for 24 h. Cells were observed using optical microscopy.

**Figure 6 pharmaceutics-14-01765-f006:**
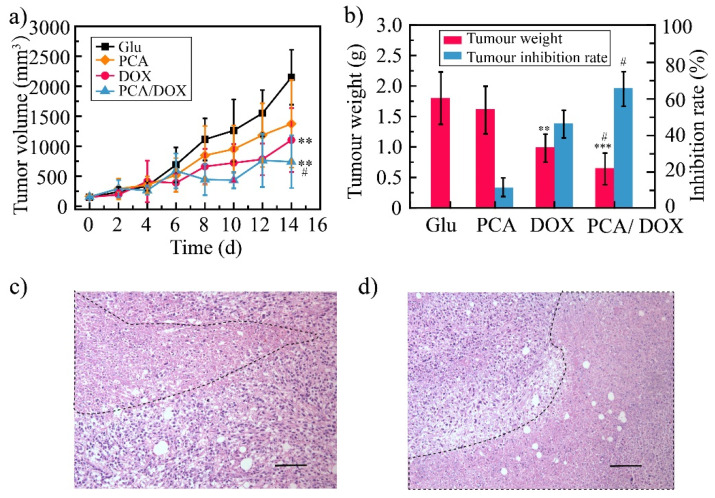
The result of antitumor efficacy: 4T1 tumor volume change curves (**a**), tumor weight and tumor inhibition rate (**b**), tumor tissue images of DOX (**c**), and PCA/DOX NPs (**d**), scale bar: 100 μm. *** *p* < 0.001, ** *p* < 0.01, vs. Glu group, ^#^ *p* < 0.05, vs. DOX group, n = 10.

**Figure 7 pharmaceutics-14-01765-f007:**
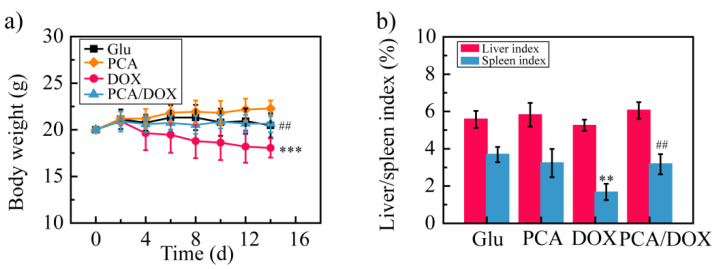
The result of DOX toxicity test: (**a**) body weight change curves; (**b**) liver and spleen index. *** *p* < 0.001, ** *p* < 0.01, vs. Glu group; ## *p* < 0.01, vs. DOX group, n = 10.

**Figure 8 pharmaceutics-14-01765-f008:**
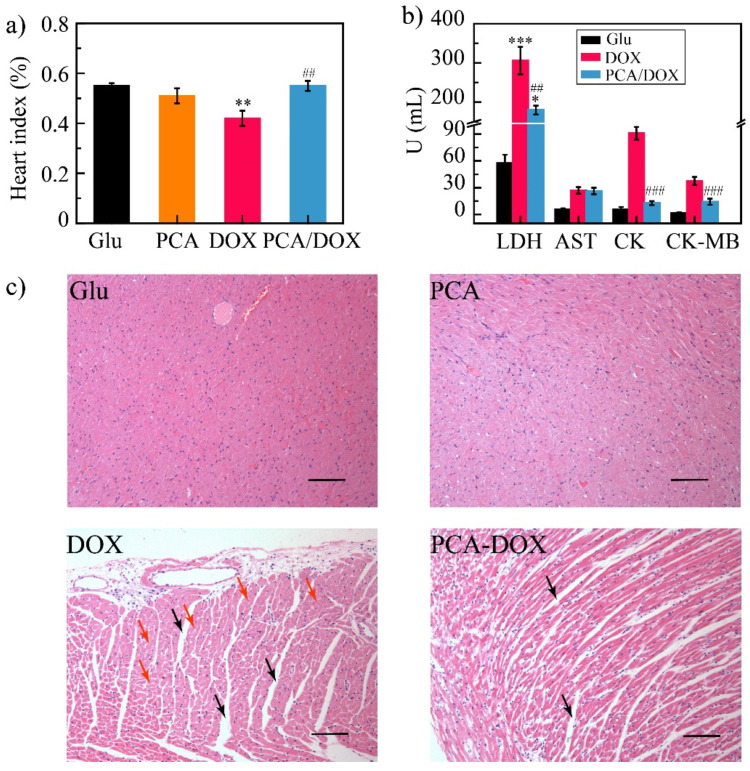
Cardiotoxicity test results: (**a**) heart index of different groups; (**b**) mice biochemical index results; (**c**) HE images—there was obvious myocardial interstitial edema in the DOX and PCA/DOX NPs groups (marked by the black arrow). The cardiomyocytes presented obvious nucleus lysis and apoptosis in the DOX group (marked by the red arrow), scale bar: 100 μm. *** *p* < 0.001, ** *p* < 0.01, * *p* < 0.05, vs. Glu group; ### *p* < 0.001, ## *p* < 0.01, vs. DOX group, n = 10.

## Data Availability

Not applicable.

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
