# Peer review of "Poly(methacrylate citric acid) as a Dual Functional Carrier for Tumor Therapy"

_pharmaceutics, 2022, doi:10.3390/pharmaceutics14091765_

Round 1

Reviewer 1 Report

The work described by Guo et al is interesting and has also the potential to be translated from basic research to clinical implications. There are some statements, that are not yet sufficiently convincing, and thus should be validated by other methods. English language editing is strongly advised.

Questions, comments, suggestions:

1. in the supplementary data file the "zeta potential of the drugs" table relates to the zeta potential of exactly what samples? The drug-loaded nanoparticles, right? Please clarify.

2. supplementary figure 2 figure legend does not specify which nanoparticles it relates to exactly. Please amend the figure legend.

3. Under cytotoxicity there are various figures shown (cell viability, cell inhibition rate) on different cell lines with different treatment conditions (within the manuscript as well as in the supplementary file), however, based on the description of the tests under Materials and methods, it is not entirely clear how the different figures related to these tests were obtained. Please provide a more detailed description.

4. Vascular growth inhibition: I think 2 representative images are not sufficient to prove this point. Provide more quantitative data and statistical evaluation to prove the statement. 

5. Similarly, to validate apoptosis the authors should present more results, obtained by different techniques (immunohistochemistry or Western blotting for apoptosis marker/s), showing one HE image from one sample and another image from another single sample from another treatment group are not sufficient. These should be provided to validate the point of apoptosis.

Author Response

Thank you very much for kind comment, a point-by-point response please see the attachment.

Reviewer 2 Report

The article is of interest and may be suitable for publication after taking into account the following remarks:

1. In the introduction, the advantage of nanosystems is the increased solubility of the drug, but in the case of doxorubicin (which is discussed in the manuscript) there is no problem with solubility and, on the contrary, it is necessary to control its distribution in the body and the rate of release. However, this circumstance is not noted in the manuscript. The following manuscripts (doi 10.1002/macp.202200081 and doi: 10.3390/polym13152569) may help here.

2. It is necessary to clearly explain the mechanism of nanoparticle formation - without this, the article looks unreasonable. 

3. In Figure 2, a trend line should be drawn rather than connecting the experimental points with a broken line.

4. Figure 3 gives release curves, but they do not reach equilibrium or full release, which does not add value to the data. Kinetic release models should be constructed for all kinetic curves.

5. Did you measure the molecular weight of the copolymer yourself or is this data from the literature? 

Author Response

(The authors gave the same response as above.)

Round 2

Reviewer 2 Report

The article can be accepted in its current form.